# 3D-Printed Acrylated Soybean Oil Scaffolds with Vitrimeric Properties Reinforced by Tellurium-Doped Bioactive Glass

**DOI:** 10.3390/polym16243614

**Published:** 2024-12-23

**Authors:** Matteo Bergoglio, Matthias Kriehuber, Bernhard Sölle, Elisabeth Rossegger, Sandra Schlögl, Ziba Najmi, Andrea Cochis, Federica Ferla, Marta Miola, Enrica Vernè, Marco Sangermano

**Affiliations:** 1Department of Applied Science and Technology, Politecnico di Torino, Corso Duca degli Abruzzi 24, 10129 Torino, Italy; matteo.bergoglio@polito.it (M.B.); marta.miola@polito.it (M.M.); enrica.verne@polito.it (E.V.); 2Polymer Competence Center Leoben GmbH, Roseggerstrasse 12, 8700 Leoben, Austria; matthias.kriehuber@pccl.at (M.K.); bernhard.soelle@pccl.at (B.S.); elisabeth.rossegger@pccl.at (E.R.); sandra.schloegl@pccl.at (S.S.); 3Department of Health Sciences, Center for Translational Research on Autoimmune and Allergic, Disease—CAAD, Università Del Piemonte Orientale (UPO), 28100 Novara, Italy; ziba.najmi@uniupo.it (Z.N.); andrea.cochis@med.uniupo.it (A.C.); 20028399@studenti.uniupo.it (F.F.)

**Keywords:** dynamic polymer networks, tellurium-doped bioactive glass, 3D-printed scaffolds

## Abstract

In this study, we present novel, vitrimeric and biobased scaffolds that are designed for hard tissue applications, composed of acrylated, epoxidized soybean oil (AESO) and reinforced with bioactive glass that is Tellurium doped (BG-Te) and BG-Te silanized, to tune the mechanical and antibacterial properties. The manufacture’s method consisted of a DLP 3D-printing method, enabling precise resolution and the possibility to manufacture a hollow and complex structure. The resin formulation was optimized with a biobased, reactive diluent to adjust the viscosity for an optimal 3D-printing process. The in vitro biological evaluation of the 3D-printed scaffolds, combined with BG-Te and BG-Te-Sil, showed that the sample’s surfaces remained safe for hBMSCs’ attachment and proliferation. The number of *S. aureus* that adhered to the BG-Te was 87% and 54% lower than on the pristine (control) and BG-Te-Sil, respectively, with the eradication of microbiofilm aggregates. This work highlights the effect of the vitrimeric polymer matrix and doped, bioactive glass in manufacturing biocompatible, biobased, and antibacterial scaffold used in hard tissue application.

## 1. Introduction

Polymeric scaffolds are essential in tissue engineering, providing a structural framework that mimics the extracellular matrix, which supports cell attachment, proliferation, and differentiation [1]. They play a critical role by delivering biochemical cues and physical support, guiding tissue growth in three dimensions. Current research emphasizes how optimizing polymeric scaffold properties for specific applications, such as bone, cartilage, and skin tissue engineering, enhances their functionality through the integration of bioactive components to improve biocompatibility, cell growth, and cell differentiation [2]. Within this frame, there is an increasing interest in the investigation of scaffolds with dynamic polymer network (DPN) properties [3].

DPNs are designed with dynamic covalent bonds, which can be activated in response to an environmental stimulus [4]. DPNs can be classified as either dissociative or associative, depending on the exchange chemistry of their dynamic bonds [5,6,7,8]. Dissociative dynamic polymer networks (DPNs) possess a balance between the formation and breaking of their crosslinks. On the counterpart, associative DPNs involve reactions that enable bond exchange within the crosslinked network. This process maintains a constant density of crosslinks while allowing changes in the network topology. As a result, the viscosity of associative DPNs changes gradually with temperature in an Arrhenius-like manner.

As mentioned, dynamic covalent bonds can be activated by an appropriate external stimulus, inducing the possibility to undergo bond breaking and reforming. This behavior can be particularly interesting for scaffolds, as it enables properties such as self-healing [9], reprocessability [10], and solid-state plasticity [11].

Fang et al. exploited heating-induced dynamic covalent bonds to achieve self-healing abilities in polymeric scaffolds [12]. Krishrakumar et al. used a biobased scaffold with covalent adaptable network properties based on amine and bicyclic carbonate precursors, showing good shape plasticity [13]. DPNs have also been exploited to activate shape memory behavior [14]. Shape memory polymers that respond to heat are useful in biomedical devices, including vascular stents [15], occlusion systems [16], and tissue scaffolds [17].

Biobased precursors are increasingly utilized when designing 3D-printed composite scaffolds for biomedical applications. These sustainable materials also offer biodegradability, making them ideal for tissue engineering. Recent studies highlight the integration of natural polymers, such as chitosan [18,19], gelatin [20], and cellulose [18,21,22], into 3D-printed scaffolds to enhance cell attachment, proliferation, and tissue regeneration. This approach not only supports eco-friendly practices, but also advances the functionality and adaptability of biomedical scaffolds [23]. In our previous paper, [24] acrylated, epoxidized soybean oil (AESO) was exploited as a biobased precursor, together with isobornyl acrylate (IBOA) as a reactive diluent, to prepare photocurable formulations containing bioactive glass particles. The investigated formulations showed high reactivity towards radical photopolymerization, and rheological studies demonstrated the feasibility of using these formulations in a 3D-printing process. The 3D-printed scaffolds demonstrated excellent cytocompatibility with human cells. Our findings showed that incorporating bioactive glass enhanced the mechanical properties of the scaffold while simultaneously improving biological outcomes, such as cell growth and proliferation. In the literature, it is reported that ions released from bioactive glasses, such as Na^+^, K^+^, and Ca^2+^, can help neutralize acidity resulting from polymer degradation [25]. Additionally, bioactive glasses can improve cell attachment and proliferation on scaffolds and stimulate bone formation by releasing Si ions [26].

To pursue this research line, we prepared tellurium-doped bioactive glass and dispersed it into 3D-printable biobased formulations. The tellurium-doped bioactive glass was selected because of several potential benefits, particularly in the context of tissue engineering and regenerative medicine, such as antibacterial properties as well as antioxidant properties [27,28,29]. Moreover, a eugenol-based transesterification catalyst and eugenol-based reactive diluent were used to underline the difference from previous work [24]. The curing process and printability of AESO-based formulations were fully characterized, as well as the printed scaffold. In particular, dynamic covalent network properties were evaluated using stress-relaxation analysis.

## 2. Materials and Methods

### 2.1. Materials

The epoxidized, acrylated soybean oil (AESO) was obtained from Allnex (Frankfurt, Germany). Phenylbis (2,4,6-trimethylbenzoyl) phosphine oxide, ammonium hydroxide (NH_4_OH), tetraethyl orthosilicate (TEOS), triethyl phosphate (TEP), and calcium nitrate tetrahydrate (Ca(NO_3_)_2_ 4H_2_O) were purchased from Sigma-Aldrich, Milano, Italy.

The acrylated eugenol (AEUG as the reactive diluent) and the eugenol-based phospate ester (EUGP as transesterification catalyst) were synthesized according to the literature [30,31]. Te-doped bioactive glass (BG-Te) was synthesized via the sol-gel process. The chemical structures of the monomer, diluent, and transesterification catalyst are reported in Table 1.

### 2.2. Bioactive Glass Synthesis

The bioactive glass (BG) was synthesized using the sol-gel method, specifically following the modified Stöber method, as described by El-Rashidy et al. [32], which was optimized in our previous research [24].

The synthesis involved mixing two solutions: one containing ethanol (EtOH) and tetraethyl orthosilicate (TEOS), and the other containing 28% ammonium hydroxide (NH_4_OH) to act as a gelation agent. Mixing these solutions induced basic hydrolysis, leading to the formation of a silica network. After this network formed, triethyl phosphate (TEP) as a phosphorus precursor and calcium nitrate tetrahydrate (Ca(NO_3_)_2_·4H_2_O) as a calcium precursor were added to the emulsion. Only after this step was the tellurium precursor, sodium tellurite (Na_2_TeO_3_), introduced to dope the glass.

Following the synthesis, the BG was dried for 48 h to remove excess water, and then calcinated in a furnace at 700 °C for two hours to remove residual nitrates and to stabilize the system. The silanization of the bioactive glass was performed by immersing the synthetized Te-doped glass into a solution of 20% 3-(Trimethoxysilyl)propyl methacrylate (TMSPMA) in ethanol, and then vigorously stirring for two hours. To ensure a complete interaction between the particles and the silanization solution, the surface of BG was previously treated via immersion in acetone in an ultrasound bath and centrifugated at 7000 rpm for 2 min to remove the acetone. FTIR and contact angle checked the occurrence of the silanization. Briefly, with the FTIR, we ensured that the typical peaks of TMSPMA were present in the FTIR spectra of silanized BG. Moreover, it was checked the change in hydrophilicity since the pristine BG was completely hydrophilic (0°), while the silanized glass was hydrophobic (around 90°).

### 2.3. Acrylated Eugenol (AEUG) Synthesis

A total of 40 mL of dry THF (stabilized) was placed in a three-necked round-bottomed flask. The whole synthesis was carried out inside an N_2_ atmosphere. Eugenol (20.04 g; 0.122 mol; 1 eq.) and Et_3_N (12.33 g; 0.122 mol; 1 eq.) were added and stirred for 10 min. Acryloyl chloride (14.36 g; 0.159; 1.3 eq.) was diluted with 10 mL of dry THF and placed in a dropping funnel. Subsequently, the solution was added dropwise over a time period of half an hour. The mixture was stirred at room temperature overnight. The formed precipitate was filtered off and the solvent was evaporated (yield 98%). The NMR spectrum was in accordance with the literature [30].

^1^ H NMR (300 MHz, CDCl_3_): δ [ppm] = 6.98 (d, J = 8.16 Hz, 1H); 6.84–6.71 (m, 2H); 6.61 (dd, J = 17.34 Hz, 1H); 6.3 (dd, J = 17.34 Hz, 1H); 6.03–5.85 (m, 2H); 5.16–5.01 (m, 2H); 3.77 (s, 3H); 3.37 (d, J = 6.64 Hz, 2H).

### 2.4. Eugenol Phosphate Ester (EUGP) Synthesis

An amount of 60 mL of toluene and POCl_3_ (7.86 g; 0.0513 mol; 1 eq.) were put in a 250 mL three-necked round-bottomed flask equipped with a reflux condenser and a thermometer. Subsequently, eugenol (15.09 g; 0.092 mol; 1.8 equiv.) and Et_3_N (10.44 g; 0.103 mol) were placed in a dropping funnel and diluted with 20 mL of toluene. The reaction mixture was slowly added dropwise into the solution under vigorous stirring over a time period of 4 h. The temperature was kept below 30 °C. The formed insoluble salt was filtered off. The reaction mixture was poured into a two-necked round-bottomed flask, and steam from an external steam generator was bubbled into the mixture for 25 min. The phases were separated with a separatory funnel and the organic layer was collected and dried over sodium sulfate. The solvent was removed using a rotary evaporator and a yellowish oily liquid was obtained (yield 95%). The NMR spectrum was in accordance with the literature [30].

^1^ H NMR (300 MHz, CDCl_3_): δ [ppm] = 7.30–7.23 (m, 1H); 6.79 (s, 1H); 6.76–6.70 (m, 1H); 6.01–5.84 (m, 1H); 5.14–5.03 (m, 2H); 3.84 (s, 3H); 3.4–3.3 (m, 2H).

### 2.5. Formulation, Photocuring, and 3D-Printing

Following our previous work, the formulation consisted of a 70:30 ratio of AESO and AEUG. AEUG was used as a reactive diluent to decrease the viscosity and to obtain a printable formulation. The resin was added to 30 phr of Te-doped bioactive glass and 2 phr (Parts per Hundred parts Resin) of phenylbis (2,4,6-trimethylbenzoyl) phosphine oxide as a photoinitiator. The formulation obtained underwent an ultrasonic bath at 50 °C to ensure complete dissolution of the photoinitiator, followed by mixing in a planetary mixer (Thinky Mixer ARE-250, Tokyo, Japan) to ensure the uniform dispersion of the bioactive glass within the resin.

The 3D-printing process was conducted using a Prusa SL1S SPEED (Czech Republic) equipped with a 405 nm UV light. The following parameters were used to conduct the printing: the 1st layer received a UV-light exposition of 120 s to ensure a strong adhesion to the metal building platform, followed by a 5 s exposure for the subsequent layers. After printing, the samples were immersed in an ultrasonic bath with isopropanol for 15 min to completely remove any excess resin and were post-cured for 30 min using a Phrozen cure kit (Taipei, Taiwan).

### 2.6. Characterization

#### 2.6.1. PhotoDSC

The photocrosslinking reaction was studied using a PhotoDSC technique. The measurement was performed by means of a Mettler Toledo DSC-1 system (Milano, Italy) that was equipped with a Hamamatsu Lightingcure LC8 UV lamp and a gas controller. The lamp was connected to an optical fiber to emit UV radiation, which was focused on the sample with a 365 nm wavelength and 50 mW/cm^2^ intensity. The sample’s weight, placed inside the chamber in an aluminum pan, varied from 5 to 15 mg. An empty aluminum pan was used as a reference. Each experiment was conducted under 40 mL/min of nitrogen flow and a constant room temperature. The analysis method consisted of an initial settling time of two minutes, a first irradiation step of 60 s, an additional two minutes of settling, and a final irradiation step of 60 s. To avoid any peaks not related to the photocuring, the second curve data were subtracted from the first. The data analysis was performed using Mettler Toledo STARe software.

#### 2.6.2. Fourier Transform Infrared Spectroscopy (FTIR)

To monitor in real-time the crosslinking reaction, FTIR measurements were performed by means of a Nicolet iS 50 Spectrometer equipped with a Hamamatsu Lightingcure LC8 lamp, with a 365 nm UV light source, at an intensity of 50 mW/cm^2^. The liquid resin was spread onto the silicon wafer substrate by means of a stir bar that allowed us to obtain a sample thickness of 32 μm. The spectra resolution was set to 4 cm^−1^, and all the data were processed with Thermo Fisher Scientific’s OMNIC software.

Conversion data were recorded by tracking the characteristic acrylate peak situated at 1640 and 1615 cm^−1^. The peak located at 2950 cm^−1^, corresponding to C-H stretching, was used as a reference, as it remained unaffected by UV light exposure. The conversion was calculated using the following Equation (1):(1)$$Conversion\;\;\left(\%\right)=\frac{{\left(\frac{A_{group}}{A_{ref}}\right)}_{t=0}-{\left(\frac{A_{group}}{A_{ref}}\right)}_{t}}{{\left(\frac{A_{group}}{A_{ref}}\right)}_{t=0}}*100$$
where A_group_ represents the acrylate peak area during the time, and represents A_ref_ the C-H peak area. The study tracked the values over the exposure time to build the conversion curve.

#### 2.6.3. Photorheology and Viscosity Measurements

The UV-curing process was also investigated using an Anton Paar MC 302 instrument. A Hamamatsu Lightingcure LC8 lamp, coupled with an optical fiber, served as a 365 nm UV light source, with an intensity of 50 mW/cm^2^. Tests were conducted using parallel plate-plate geometry, where the upper metal plate had a 2.5 cm diameter, and the lower plate consisted of a quartz disk to allow direct irradiation of the sample. The distance between the crystal and the plate corresponded to 100 μm. The frequency was set to 1 Hz with a 1% strain. The Hamamatsu lamp was activated after 60 s to ensure the stability of the system before the start of the measurement.

The same Anton Paar MC 302 instrument was used with a different setup to determine the viscosity of the different formulations. The test involved plate-plate parallel geometry using both metal plates. The shear rate ranged from 0.1 to 1000 s^−1^ during the test.

#### 2.6.4. Dynamic Mechanical and Thermal Analysis (DMTA)

DMTA measurement was conducted using a Triton Technology instrument to examine the change in the storage and loss modulus, and consequently the tan δ, by varying the temperature. The sample dimension was 1 × 8 × 18 mm. The test started from 0 °C, which was reached by cooling the chamber with liquid nitrogen. Measurements were performed in tensile mode, applying a 1 Hz frequency and an initial displacement of 0.02 mm. The measurement was stopped when the storage modulus reached the rubbery plateau. Glass transition temperature (T_g_) was determined at the maximum of the tan δ curve.

#### 2.6.5. Mechanical Tests

Stress-strain measurements were performed at room temperature using an MTS QTest™/10 Elite electromechanical universal testing machine (MTS System Corporation, Eden Prairie, MN, USA), operated through TestWorks^®^ 4 software. A 500 kN load cell was employed, and the machine’s crosshead speed was set to 5 mm/min. The samples tested followed the ISO-527A-5B shape. Young’s modulus (E) was calculated as the initial slope in the linear range of the curve. The values reported represent the average values of five individual experiments.

#### 2.6.6. FESEM

The morphological analysis of the samples’ fracture surface was conducted to examine the particle dispersion and their interaction with the polymer matrix, and to investigate the formation of the hydroxyapatite (HAp) onto the sample surface after immersion in simulated body fluid (SBF), performed using the Kokubo’s protocol [33]. The samples were mounted on stubs and coated with a 5 nm Platinum layer to improve conductivity. The Zeiss Supra 40 FESEM was used to perform the analysis.

#### 2.6.7. Stress Relaxation Measurement

Stress relaxation experiments were performed using a Physica MCR 501 rheometer (Anton Paar, Graz, Austria). Samples, with dimensions of 0.5 mm in thickness and 10 mm in diameter, were conditioned at a constant temperature for 15 min. Afterwards, a constant strain of 3% was applied, and the stress release was recorded over time. Testing temperatures ranged from 180 to 220 °C.

The relaxation modulus G(t) was normalized to the initial modulus G_0_, measured at the onset of strain application. Relaxation time, a key property of DPNs, was defined as the time required for the relaxation modulus to decrease to 1/e of its initial value, exhibiting an exponential decay as described by Equation (2).
(2)$$G\left(t\right)=G_{t0}e^{\left(\frac{-t}{\tau }\right)}$$

### 2.7. Biological Evaluation of 3D-Printing Samples

#### 2.7.1. Direct Cytocompatibility Assessment

To evaluate the cytocompatibility of 3D-printed scaffolds containing Te and silanized Te-doped BG, human bone marrow cell-derived mesenchymal stem cells (hBMSCs, purchased from PromoCell, C-12974) were chosen as a cell model for tissue regeneration application, particularly bone regeneration. The hBMSCs were seeded in Dulbecco’s modified eagle medium (DMEM, Thermo Fisher, Milan, Italy), supplemented with 15% Fetal bovine serum (FBS, Sigma Alderich, Milan, Italy) and 1% antibiotics, including penicillin and streptomycin. They were incubated at 37 °C with 5% CO_2_.

In direct contact evaluation, following the International Organization for Standardization (ISO) protocol (ISO10993-5), hBMSCs (15,000 cells) were seeded directly on each sample and incubated for 4 h to allow cell adhesion to the sample surfaces. Then, 500 µL of fresh culture medium was added. After 24 h of incubation, the cells’ metabolic activity, viability, and morphology were analyzed using colorimetric Alamar blue^TM^ assay, double fluorescent staining with live/dead reagent (Live/Dead Viability/Cytotoxicity Kit for mammalian cells, L3224, Invitrogen, Milan, Italy) and NucBlue^TM^ Live Cell Stain ReadyProbes^TM^ reagent (R37605, Invitrogen, Milan, Italy), and scanning electron microscopy (SEM, JEOL, Tokyo, Japan), respectively, as explained in a previous article [24]. The results of metabolic activity, measured by the absorbance of metabolized Alamar blue reagent, are reported as relative fluorescent unit (RFU) values. Additionally, to evaluate the cytotoxic effects of the 3D-printed scaffolds on hBMSC cells, the release of the cytosolic enzyme lactate dehydrogenase (LDH) was measured using a CyQUANT^TM^ LDH cytotoxicity assay-fluorescent kit (C20303, Invitrogen, Milan, Italy); according to the mentioned formula in the manufacturers’ instructions, the cytotoxicity (%) and viability (%) were calculated:(3)$$Cytotoxicity\;\left(\%\right)=\left[\frac{compound\;treated\;LDH-Spontaneaus\;LDH}{Maximum\;LDH-Spontaneaus\;LDH}\right]\times 100$$
(4)Viability (%)=1−Cytotoxicity (%)

#### 2.7.2. Antibacterial Assessment

As the study aims to enhance the tissue regeneration capability, particularly for bone, of 3D-printed scaffolds composed of BG with Te and silanized Te, *Staphylococcus aureus*, a Gram-positive bacterium that is a primary concern in orthopedic infections [34], was selected for antibacterial activity evaluation. *S. aureus* was purchased from the American-type culture collection (ATCC, ATCC 43300, Manassas, VA, USA), and according to the manufacturer’s instructions, Lauria Bertani (LB) broth/agar was used as the culture medium for bacterial growth at 37 °C.

Before the experiment, a fresh subculture was prepared. After allowing bacterial growth, optical density was measured at 600 nm (OD600nm). The desired bacterial concentration (1 × 105 Colony forming unit (CFU)/mL) for the experiment was achieved by diluting the culture with fresh culture medium. To evaluate the direct antibacterial effect of the samples on the bacteria, 100 µL of bacterial suspension was directly applied to each sample’s surface and incubated for 24 h at 37 °C. After incubation, the samples were washed briefly with Phosphate-buffered saline (PBS) to remove non-adherent bacterial cells. As described in a previous article [35], metabolic activity and the viable number of bacterial colonies attached to the sample surfaces were analyzed using the colorimetric Alamar Blue assay and CFU counting, respectively. Finally, the morphology of surface-adhered bacterial cells and aggregates was observed using SEM [35]. SMILE View^TM^ Map 8.2.9621 software (JEOL, Tokyo, Japan) was used for post-modifications of SEM images, in which the bacterial cells and aggregates were colored to be distinguishable from the BG particles.

### 2.8. Statistical Analysis

Statistical analysis of the obtained results was conducted using the SPSS software (v.20.0, IBM, Armonk, NY, USA). The normal distribution and homogeneity of variance of the data were initially confirmed by Shapiro–Wilk’s and Levene’s tests, respectively. Differences between groups were then analyzed using the one-way ANOVA, and using Tukey’s test as the post-hoc analysis. Significant differences were established at *p* < 0.05 and indicated here by an asterisk (*).

## 3. Results and Discussion

Pursuing a previous study on 3D printing a biobased scaffold reinforced with bioactive glass [24], we have now examined the reactivity of an acrylate-based formulation derived from soybean oil, incorporating eugenol acrylate as a biobased reactive diluent. The bioactive glass (BG) was synthesized on purpose through a low-energy consuming sol-gel synthesis. Furthermore, it was doped with tellurium ions, as reported in the experimental part, and eventually silanized to improve filler-matrix interactions.

As reported by Nguyen et al., silanization, particularly using TMSPMA ((3-(Trimethoxysilyl)propyl methacrylate), is an effective method to enhance the dispersion of particles within a polymer matrix, reducing their tendency to aggregate, thereby improving the final mechanical properties of the final composite [36,37]. Among various silanization techniques, the use of TMSPMA emerges as an efficient and simple option. This method saves time and energy and avoids the use of xylene as a solvent, relying only on ethanol. The silanization process involves the formation of Si-O-Si bonds on the particle’s surface. Additionally, the methacrylate groups present in TMSPMA can participate in the photocuring process, further improving the integration of nanoparticles into the polymer matrix.

A fixed content of BG was selected on the basis of a previous investigation, which showed a 30 phr BG content as an optimum amount to induce bioactive properties to the scaffold [24]. This was confirmed for scaffolds containing BG doped with tellurium ions, as reported below. Both acrylated eugenol (AEUG), which has been exploited as a reactive diluent during the 3D-printing process, as well as the eugenol-based phosphate ester (EUGP), used as a transesterification catalyst, were synthesized as reported in the experimental section, with the aim of designing complete biobased photocurable formulations.

### 3.1. Investigation of Photocuring Process

The reactivity of AESO-based formulations was investigated using FTIR analysis. The investigated formulations are reported in Table 2. The pristine formulation is composed of AESO and AEUG in a 70:30 ratio. This ratio between AESO and the reactive diluent was determined based on findings from a previous investigation [24]. The filled formulations contained 30 phr of either BG doped with tellurium (BG-Te) or the silanized BG doped with tellurium (BG-Te-Sil). All the formulations contained 10 phr of EUGP as a biobased transesterification catalyst. The appropriate amount of catalyst in order to achieve dynamic covalent networks was determined using a preliminary stress-relaxation test not reported in this study.

Conversion curves as a function of irradiation time and typical FTIR spectra are reported in Figure 1 for the investigated formulations. The acrylic carbon double bond conversion band centered at 1640 cm^−1^ was followed during irradiation and the final conversion after 2 min is shown in Table 2.

From the curves reported in Figure 1 and the data collected in Table 2, it is possible to observe a decrease in the conversion rate and final carbon double bond conversion in the presence of the doped BG. When the pristine formulation reached an overall double bond conversion of about 86% after 2 min of irradiation, the filled formulation containing the tellurium-doped BG reached 80% of conversion, and the silanized tellurium-doped BG reached only 54%.

The detrimental effect of the presence of the filler in the curing process could be attributed to a competitive absorption effect of light between the glass and the photoinitiator, with a decrease of photoinduced reactive species as well as to a light scattering effect due to the micrometric dimension of the fillers.

Photo-DSC analyses were also performed on the same formulations. In Figure 2, the heat flow as a function of irradiation time is reported for the pristine and for the BG-Te and BG-Te-Sil photocurable formulations. The integral of the heat flows, which is related to the acrylic double bond conversion, with a higher polymerization heat through increasing double bond conversion, is collected in Table 2 together with the time to reach the maximum peak.

The photo-DSC data agree with the FTIR data, showing a significant decrease in acrylic double bond conversion by adding the filler in the photocurable formulation, with a heat-flow for the pristine formulation of about 215 J/g, which decreased to 175 J/g for the BG-Te and to 189 J/g for the BG-Te-Sil formulation, respectively. Additionally, the increase in time to reach the peak of the heat flow for the filled formulations indicates a slower photocuring process when the BG is dispersed. Comparing BG-Te and BG-Te-Sil, the heat released for the formulation containing silanized particles is approximately the same as BG-Te, despite the different behaviour observed in FTIR conversion. This is attributed to the methacrylic groups contained in the silanizing agent. These groups, although reactive, are not visible in the FTIR conversion, while during a photo DSC experiment, the heat released shows no distinction between different contributions.

To confirm the occurrence of an optimal photocuring process, photorheology was performed. The trials, reported in Figure 3, show how the modulus G’ increases when the light is switched on, confirming once more the photoinitiator role of the pristine resin. All the tested formulations show a fast reactivity, reaching the plateau after approximately 20 s of irradiation.

Notwithstanding the lower acrylic double bond conversion upon curing for the filled formulations, it was always possible to achieve a tack-free crosslinked material.

### 3.2. Scaffold 3D-Printing

The rheology of photocurable formulations was tested in order to define their printability. In Figure 4, the viscosity vs. shear rate curves are reported for the pristine and for the BG-Te and BG-Te-Sil formulations. It is possible to observe that the formulations present a Newtonian behavior (constant viscosity as the shear rate increases) in the considered shear rate range. The viscosity of the formulations is in the range of printability [38,39].

After evaluating the suitable viscosity for printability, 3D-printing parameters were defined for all the investigated formulations, which corresponded to 120 s for the first layer and 5 s for the subsequent layers. On the basis of these parameters, all the formulations were 3D-printed. Both DMTA rectangular specimens, as well as 3D-complex structures, were achieved using 3D printing for both pristine and filled formulations. In Figure 5, an example of a complex shape obtained with BG-Te is reported.

### 3.3. Characterization of 3D-Printed Formulations

#### 3.3.1. Dynamic Mechanical and Thermal Analysis

The 3D-printed formulations were characterized using DMTA to analyze the thermo-mechanical properties of the achieved networks. The storage modulus and tan δ curves are reported for the 3D-printed formulations in Figure 6. The maximum of the tan δ curve is defined as the T_g_ of the crosslinked network. While the T_g_ of the crosslinked pristine formulation is centered around 70 °C, both the crosslinked BG-Te and BG-Te-sil showed a T_g_ of around 60 °C. The decrease in T_g_ by adding the filler can be attributed to the decrease of carbon double bond conversion measured using FTIR and photo-DSC analysis during curing. The data are shown in Table 3.

#### 3.3.2. Tensile Tests

Mechanical properties have been investigated through tensile measurements. Figure 7 shows the stress-strain curves of the 3D-printed samples. Young’s modulus (E), stress at break (σb), and strain at break (εb) for all the samples are summarized in Table 3.

The silanization process induced an improvement in the mechanical performance of the 3D-printed scaffold. This can be attributed to the enhancement of the filler-matrix interactions due to the methacrylated groups on the BG’s surface after silanization. This was evidenced through FESEM morphological analysis on the surface fracture of the analyzed samples (see Figure 8).

Since the mechanical performance of a scaffold must mimic the properties of the bone or the tissue that it has to replace, it is important to compare the mechanical properties of bone with those obtained in this study. However, estimating a single value for a bone is challenging due to its heterogeneity in density and architecture, even within the same anatomical site. For instance, the elastic modulus of a trabecular bone ranges from 10 to 3000 MPa [40]. In our case, mechanical tests were conducted on unstructured samples, and BG-Te-Sil samples demonstrated elastic modulus values near the lower limit of trabecular bone. Given that the material properties can be tuned by adjusting its porosity and density, there is potential to achieve the desired mechanical performance by modifying and studying the G-code used for 3D printing to create more complex and mechanically robust structures on demand [41]. Additionally, it is proven that in vivo testing often results in increased mechanical performance due to biological adaptation and material integration [42].

A comparison with the mechanical properties of skin also underscores the versatility of the 3D printed objects. The Young’s modulus of skin along the tibial axis ranges from 0.3 to 20 MPa, aligning with the performance of our materials, making them suitable for application also in tissue engineering [43].

The FESEM images report visually on the interaction between the BG and the matrix in both cases. For the BG-Te sample, a poor interaction between the reinforcing phase and the matrix is evidenced by the voids present on the surface, meaning a pull-out of the particles during the fragile breaking of the sample. In contrast, in the case of BG-Te-Sil particles, the voids are no longer present on the surface. Moreover, the particles are embedded in the matrix that covers them completely.

### 3.4. Bioactivity Assessment via Simulated Body Fluid (SBF) Immersion

To analyze the growth of HaP, the precursor of the human bone, onto the scaffold’s surface, the scaffold’s surface was analyzed after 28 days of soaking it in SBF solution. The FESEM image, reported in Figure 9, shows clearly the formation of the hydroxyapatite onto the surface of both the samples analyzed, the BG-Te and BG-Te-Sil. It is evident the typical HaP form is present on the surface, suggesting that the scaffold 3D-printed objects can be a suitable substrate to enhance and support human bone formation.

### 3.5. Stress Relaxation Measurements

The AESO structure, containing both esters and -OH groups, suggested the possibility of imparting dynamic covalent network properties to the UV-cured composite scaffolds. It is important to disperse a transesterification catalyst in the crosslinked material, which can be exploited for reversible transesterification reactions.

The dynamic covalent network properties induced by the thermo-activated bond exchange reactions and the related change in the viscosity were evaluated using stress relaxation experiments, in which the decreasing force, needed to deform the material under a constant strain, is followed over time.

The normalized stress-relaxation modulus decrease can be ascribed by an exponential behavior (Equation (2), where *τ* represents the experimentally measured relaxation time at which (G(t)/G_0_) equals *1/e* (the value described by Leibler of 10^12^ Pa s) [44,45].

To evidence the Arrhenius relationship and, therefore, a dynamic associative exchange mechanism, the plot of ln*τ* as a function of 1/T should be a straight line.

As a preliminary determination, the stress-relaxation tests were performed on 3D-printed scaffolds containing 30 phr of BG with different contents of a transesterification catalyst. The best content of EUGP was determined to be 10 phr (curves not reported). Finally, a stress-relaxation test was recorded for the 3D-printed scaffold at different temperatures up to 220 °C, defined as the highest temperature below the thermal degradation of the material. The stress-relaxation curves are reported in Figure 10, together with the relative Arrhenius plot for the 3D-printed composite scaffold achieved from BG-Te formulation. The 3D-printed scaffold reached 1/e relaxation modulus already at 180 °C, decreasing relaxation time by increasing temperature. Relaxation time as a function of 1/T shows a double linear fitting, as already obtained in a vitrimeric formulation that we tested in a different project [46,47]. Similar behavior was recorded for the 3D-printed composite scaffold achieved from BG-Te formulation. Therefore, we can assume that we were able to obtain 3D-printed composite scaffolds characterized by dynamic covalent network properties.

### 3.6. Cytocompatibility Evaluation of the 3D-Printed Samples

Human bone marrow-derived mesenchymal stem cells (hBMSCs) are well-known for their regenerative and self-healing capabilities, making them advantageous for studying tissue regeneration [48]. Direct cytocompatibility, which involves direct contact between cells and the sample’s surface, was performed by seeding the cells directly onto the samples. After 24 h of incubation, their metabolic activity, viability, and morphology were analyzed, and the results were compared with pristine samples as a control. The results are presented in Figure 11a–c. As shown in Figure 11a, the RFU values, which indicate the metabolic activity of the hBMSCs on the sample surfaces, for BG-Te and BG-Te-Sil, were similar to the value for the control sample, with no statistically significant reduction (*p*-value > 0.05) observed between them. The LDH assay was used to evaluate the cytotoxicity of the samples on the hBMSCs by measuring the release of cytosolic LDH enzyme. According to the ISO 10993-5 protocol and LDH assay guidelines, viability (%) below 70% is considered to be toxic for cells. By calculating cytotoxicity (%) and viability (%) using Formulas (3) and (4), as mentioned in Section 2.7.1, hBMSC viability for all tested samples (pristine, BG-Te, and BG-Te-Sil) was higher than 70% (as indicated by a red line in Figure 11b). These results were confirmed by the SEM images and double fluorescent staining (live/dead for viable cells colored green, and NucBlue for cell nuclei colored blue) (Figure 11c). As shown in the SEM and fluorescent staining images, the hBMSCs were attached and well-spread on the sample surfaces, with most cells alive (no red cells indicating dead cells were observed). A previous study reported that AESO:IBOA with a 70:30 ratio, including 30 phr BG, was more cytocompatible compared to other AESO:IBOA ratios and BG concentrations [24]. In this study, consistent with previous research, combining the prior formulation with Tellurium (Te) and Te-silanized (Te-Sil) doped BG demonstrated that the samples remained safe for hBMSC cells to attach and maintain metabolic activity. 

### 3.7. Antibacterial Activity Evaluation of 3D-Printed Scaffolds

After confirming the cytocompatibility of the 3D-printed scaffolds, including the Te- and Te-Sil-doped BG, the experiments evaluated their antibacterial activity against the Gram-positive orthopedic pathogen, *S. aureus*. The experiments were conducted through direct contact between the bacterial strain and the samples’ surfaces, followed by incubation for 24 h at 37 °C. Figure 12a–c present the results of metabolic activity, viable colony counts of the surface-adhered bacteria, and their morphology. As shown in Figure 12a, the metabolic activity of the bacteria on the BG-Te was lower than that on the pristine (as a control) and the BG-Te-Sil, although this reduction was not statistically significant (*p*-value > 0.05). Additionally, viable surface-adhered bacterial colonies, calculated as CFU counts, showed that the number of bacteria on the BG-Te surface was about 87% and 54% fewer than on the control and the BG-Te-Sil surfaces, respectively (Figure 12b). Examination of bacterial cell morphology and aggregates demonstrated that the bacteria on the pristine surface (control) and the BG-Te-Sil formed micro-biofilms. In contrast, on the BG-Te surface, the biofilm scattered, and only single bacterial cells were observed (Figure 12c). Bacterial cells in the SEM images were colored orange using SMILE VIEW^TM^ software to distinguish them from the BG particles, which range in size from 400–570 nm [24] in comparison to 0.5–1.5 µm for *S. aureus.* This observation of antibacterial activity and inhibition of biofilm formation is confirmed by previous literature, which reported similar effects of Te-containing bioactive glass against two Gram-positive bacteria, *S. aureus* and *S. epidermidis,* under dose-dependent conditions [28]. As shown in Figure 8, after silanization, the BG particles on the surfaces were embedded and completely covered with the silanization reagent. The inefficiency of the antimicrobial activity of BG-Te-sil may be due to the silanization reagent covering the Te, which hindered the internalization of Te by bacteria and prevented its mechanism of action. In other words, after the silanization procedure, Te was less accessible to bacteria, and bacterial cells could not detect Te on the BG-Te-sil surface. Based on the study performed by Zannoni et al., the internal passing of Te through the bacterial cells and membranes can block various enzymatic reactions required for bacterial growth and biofilm formation [49].

## 4. Conclusions

The 3D-printed vitrimeric scaffold reinforced with BG-Te-doped and BG-Te-doped silanized bioglass demonstrates a remarkable combination of biocompatibility, antibacterial activity, and mechanical performance, making it a strong candidate for hard tissue applications. Furthermore, its synthesis from natural resources and the use of UV light—a low-energy method compared to traditional thermal processes—makes this new resin composition particularly noteworthy within a circular economy framework. The reprocessability of the material, enabled by transesterification reactions, further enhances its sustainability.

The cytocomaptibility evaluation of the 3D-printed scaffolds, including the BG-Te and BG-Te-Sil towards the hBMSCs, demonstrated that the samples’ surfaces were cell-friendly, which cells not only attached to and were well-spread on, but also according to the fluorescent staining and LDH assay, the majority of cells (more than 95%) were alive. Antibacterial activity assessment showed that biofilm formation was inhibited on the surfaces of BG-Te samples, and only some scattered bacterial cells were detected. This biofilm inhibition is particularly valuable given the challenge of biofilm removal from medical devices.

Regarding mechanical properties, the scaffolds exhibited robust performance suitable for hard tissue applications. Furthermore, the mechanical properties can be tailored by modifying the formulation, as demonstrated by a 65% increase in stress at the break while maintaining constantly the strain at the break when utilizing the silanizing agent.

A possible future direction for this project involves a more detailed investigation of how scaffold geometry affects performance. Understanding and optimizing the mechanical and biological properties of the final scaffold through adjustments in density, porosity, and geometry could be highly impactful. Modifying these parameters in 3D-printed structures would potentially enhance both strength and tissue integration. This approach could lead to more customized and effective solutions for specific tissue engineering applications.

In summary, the scaffold developed in this study integrates biocompatibility, antibacterial functionality, and mechanical robustness. These properties make it an innovative material suitable for applications in hard tissue regeneration and the development of infection-resistant medical devices.

## Figures and Tables

**Figure 1 polymers-16-03614-f001:**
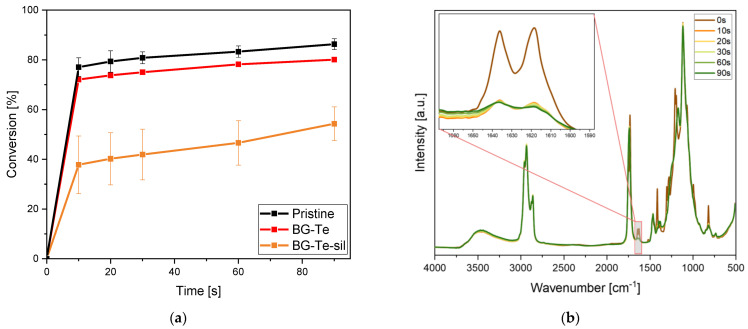
(**a**) Conversion curves as a function of irradiation time for the pristine AESO-AEUG 70:30 formulation, and for the same formulation containing 30 phr BG doped with tellurim (BG-Te) and 30 phr of silanized BG doped with tellurium (BG-Te-Sil). (**b**) Example of typical FTIR graphs (it is reported for BG-Te) at varying irradiation times. All formulations contained 2 phr of radical photoinitiator and 10 phr of EUGP as a biobased transesterification catalyst.

**Figure 2 polymers-16-03614-f002:**
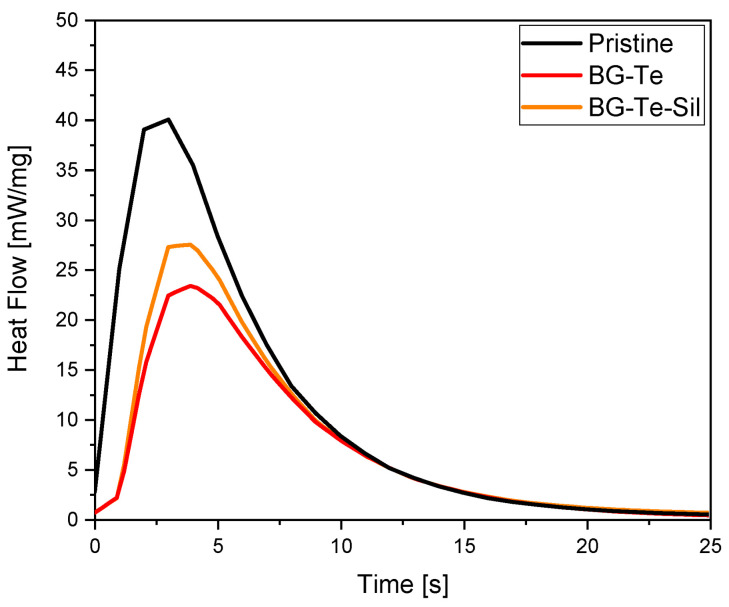
Heat flow of the photocurable pristine, BG-Te, and BG-Te-Sil formulations.

**Figure 3 polymers-16-03614-f003:**
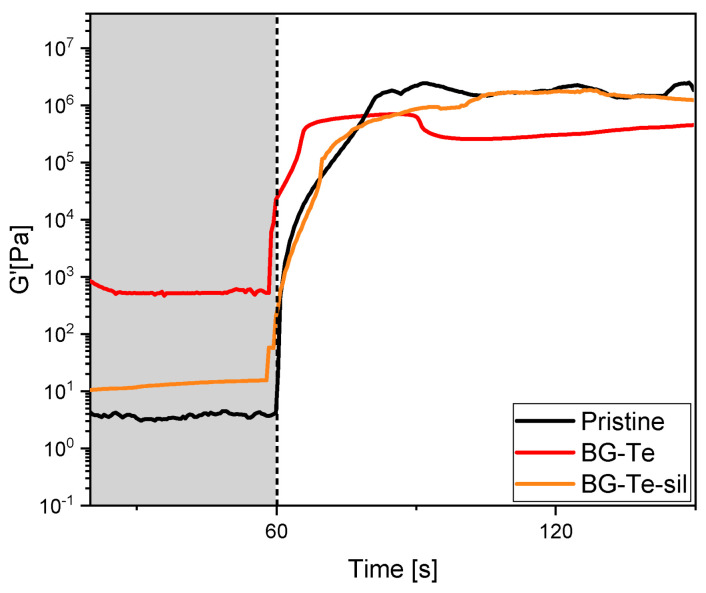
Photorheology curves of the photocurable pristine, BG-Te, and BG-Te-Sil formulations.

**Figure 4 polymers-16-03614-f004:**
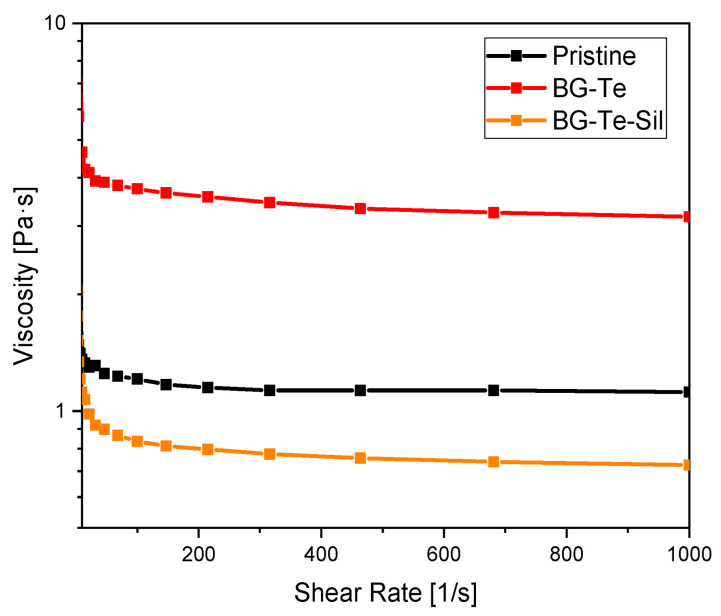
Viscosity vs. shear rate curves for the pristine and for the BG-Te and BG-Te-Sil formulations.

**Figure 5 polymers-16-03614-f005:**
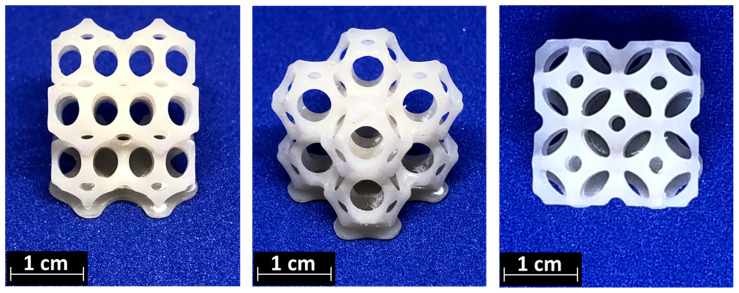
3D-printed scaffolds from different points of view.

**Figure 6 polymers-16-03614-f006:**
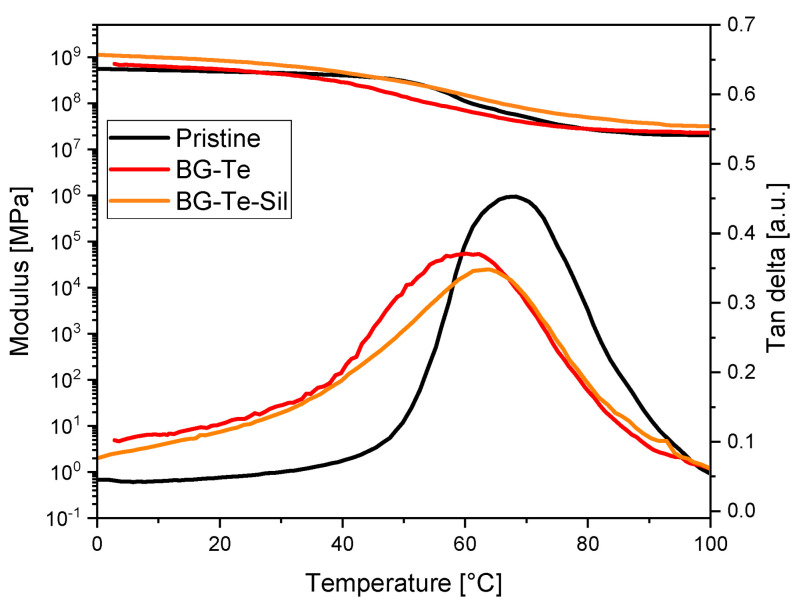
Storage modulus and tan δ curves measured for 3D-printed pristine, BG-Te, and BG-Te-Sil photocured formulations.

**Figure 7 polymers-16-03614-f007:**
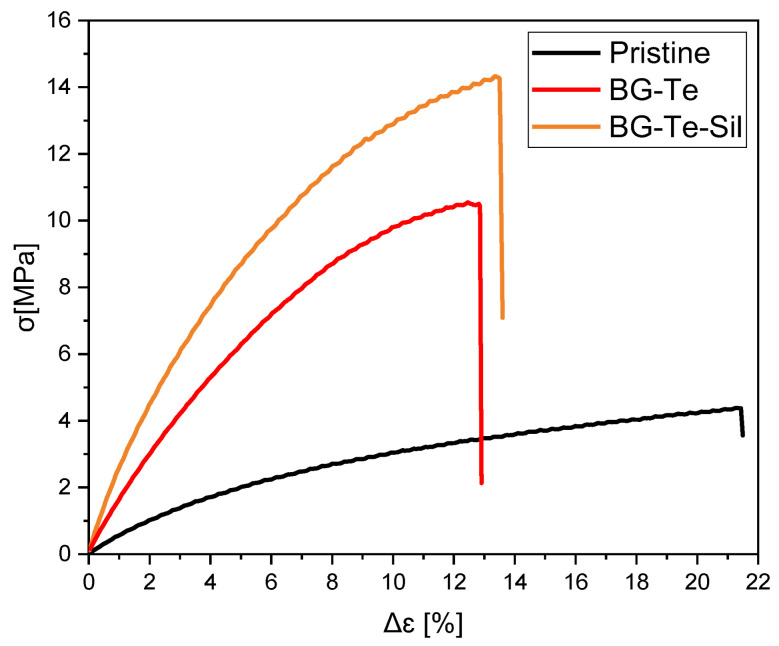
Mechanical properties of 3D-printed pristine, BG-Te, and BG-Te-Sil photocured formulations.

**Figure 8 polymers-16-03614-f008:**
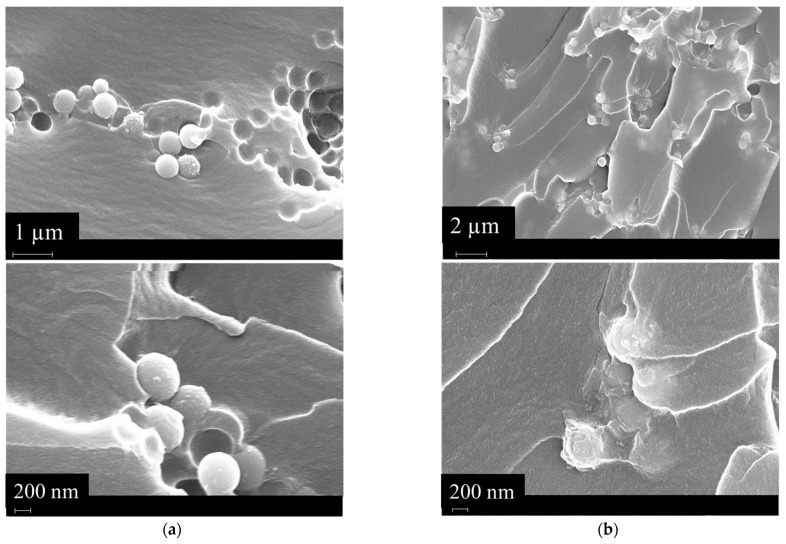
(**a**) FESEM image of the BG-Te surface and (**b**) FESEM image of the BG-Te-Sil surface.

**Figure 9 polymers-16-03614-f009:**
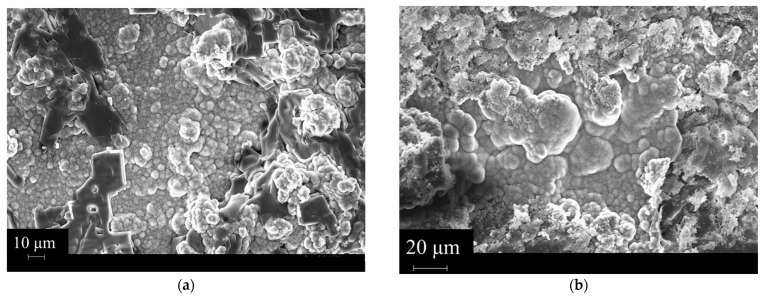
(**a**) FESEM image of the BG-Te surface after 28 days of SBF soaking and (**b**) FESEM image of the BG-Te-Sil surface after 28 days of SBF soaking.

**Figure 10 polymers-16-03614-f010:**
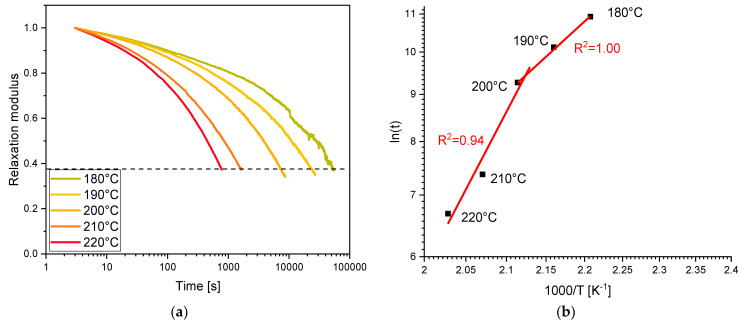
(**a**) Stress relaxation measurement of BG-Te formulation and (**b**) Arrhenius plot derived from stress relaxation measurement.

**Figure 11 polymers-16-03614-f011:**
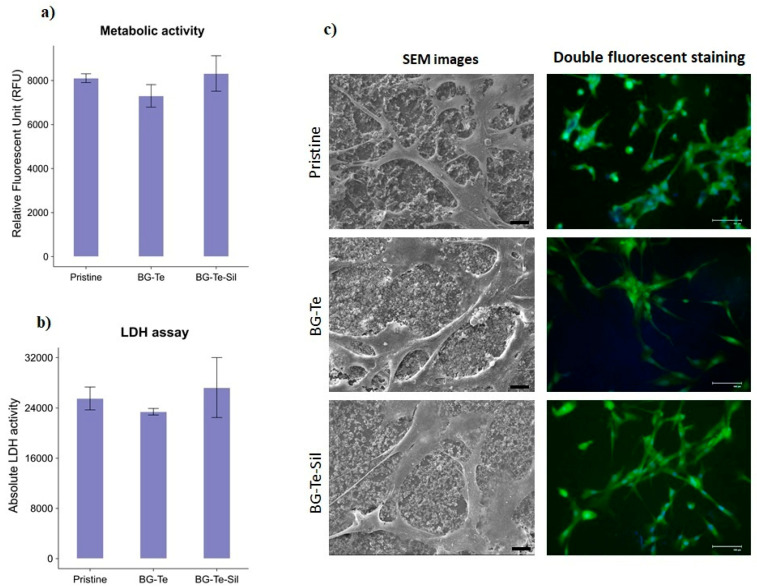
Direct cytocompatibility evaluation of the 3D-printed scaffolds towards hBMSC cells after 24 h of incubation. (**a**) Metabolic activity of the cells attached to the surfaces; (**b**) LDH assay: the red line indicates the threshold of the cells’ viability (70%); (**c**) Left panel: SEM images, scale bar = 20 µm; Right panel: double fluorescent staining with live/dead (viable cells colored green) and NucBlue (cells’ nuclei colored blue) reagents, scale bar = 100 µm.

**Figure 12 polymers-16-03614-f012:**
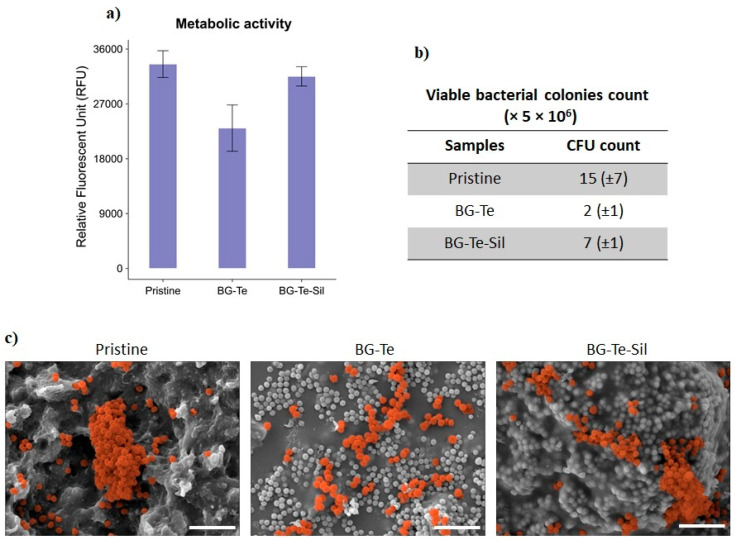
Antibacterial activity evaluation of 3D-printed scaffolds against *S. aureus* after 24 h of incubation. (**a**) Metabolic activity of the surface-adhered bacteria; (**b**) Viable bacterial colonies on the samples’ surfaces; (**c**) SEM images: bacterial cells were colored orange using SMILE VIEWTM software, scale bar = 5 µm.

**Table 1 polymers-16-03614-t001:** Chemical structure of the biobased chemicals utilized in the resin.

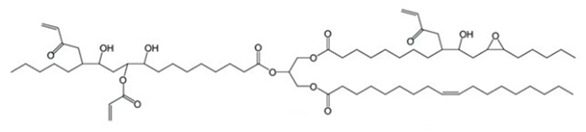	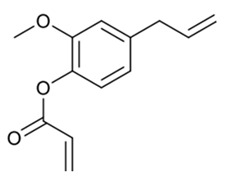	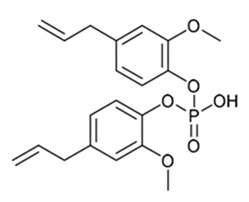
Epoxidized, acrylated soybean oil (**AESO**)	Acrylated eugenol (**AEUG**)	Eugenol-based phospate ester (**EUGP**)

**Table 2 polymers-16-03614-t002:** Composition of the investigated photocurable formulations, carbon double bond conversion, exothermicity of curing process, and time to reach the maximum peak height (t_max_).

Formulations	Conversion [%] ^1^	Heat Flow [J/g] ^2^	t_max_ [s] ^3^
AESO-AEUG 70:30 (**Pristine**)	86 ± 2	215.4 ± 16.4	3.5 ± 0.7
Pristine + 30 phr BG-Te (**BG-Te**)	80 ± 0.3	175.5 ± 17.5	4.5 ± 0.0
Pristine + 30 phr BG-Te-Sil (**BG-Te-Sil**)	54 ± 6.8	189.5 ± 1.2	4.0 ± 0.0

^1^: measured by FTIR analysis; ^2^: measured by Photo-DSC, ^3^: measured by Photo-DSC.

**Table 3 polymers-16-03614-t003:** Values of glass transition temperature (T_g_) and mechanical properties.

Formulations	T_g_ [°C] ^1^	Young’s Modulus (E) [MPa] ^2^	Stress at Break (σb) [MPa] ^2^	Strain at Break (εb) [%] ^2^
(**Pristine**)	68 ± 2	0.5 ± 0.1	4.8 ± 0.8	28 ± 5
(**BG-Te**)	61 ± 1	1.4 ± 0.0	10.5 ± 0.0	10 ± 0
(**BG-Te-sil**)	64 ± 1	1.5 ± 0.0	14.0 ± 0.0	13 ± 2

^1^ measured as the maximum of tan δ curves from DMTA analysis; ^2^ measured using tensile stress analysis.

## Data Availability

Data are contained within the article.

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
