# Peer review of "3D-Printed Acrylated Soybean Oil Scaffolds with Vitrimeric Properties Reinforced by Tellurium-Doped Bioactive Glass"

_polymers, 2024, doi:10.3390/polym16243614_

Round 1
Reviewer 1 Report
Comments and Suggestions for Authors
Review and Critique of the Article:
"3D Printed Acrylated Soybean Oil Scaffolds with Vitrimeric Properties Reinforced by Tellurium Doped Bioactive Glass"
Â
This research article describes a novel way for creating biocompatible and antibacterial scaffolds for hard tissue applications using vitrimeric polymer matrices and tellurium-doped bioactive glasses (BG-Te). This research study has a multidisciplinary approach to material science, bioengineering, and sustainable practices.
Â
Strengths
One of the study's main merits is its clear and complete approach, which is thoroughly described in the "Materials and Methods" section. The authors describe the synthesis procedures in detailed version, including the sol-gel approach for bioactive glasses, as well as photocuring and 3D printing settings. This clarity ensures repeatability and indicates rigorous experimental controls.
Â
The study also succeeds at integrating biocompatibility, antimicrobial functionality, and mechanical performance. The findings of cytocompatibility tests using hBMSCs demonstrate the scaffold's safety and potential for tissue engineering applications. The scaffolds' practical significance in reducing orthopaedic infections is highlighted by their antibacterial activity against S. aureus, which results in notable decreases in bacterial adhesion and biofilm development.
Â
The authors also highlight sustainability by using biobased components such as acrylated epoxidised soybean oil (AESO) and UV curing, which are ecologically safe methods. The research's linkage with the circular economy is noteworthy.Â
Â
Weaknesses
Despite its positive aspects, the study might benefit from a more in-depth investigation of the limits of the silanization process. The authors point out that silanization may limit acrylic double bond conversion due to interference from methacrylic groups. However, a more detailed description of alternative methodologies or changes to address this issue would improve the study's practical application.
Â
Furthermore, while the mechanical characteristics of the scaffolds are reviewed, the study may benefit from a comparison with other well-established materials used in comparable applications. Including such data might enhance the case for scaffold superiority or offer context for their performance.
Â
Finally, the display of some data, such as in Figure 7 (mechanical characteristics), might benefit from the addition of error bars to express variability and increase credibility.
Â
Suggestions
1.    While the paper describes the silanization process and its effect (e.g., hydrophilicity changes, FTIR analysis), it does not explicitly state why silanization was necessary or how it contributes to the overall performance of the bioactive glass in the polymer matrix.
2.    To overcome noted limitations, the authors may investigate alternate silanization agents or procedures that lessen negative impacts on polymer conversion rates.
3.    The paper does not reference whether this specific silanization approach has been employed in prior studies or if it was developed specifically for this research. The choice of TMSPMA as a silanizing agent is appropriate due to its methacrylate groups, which can participate in the photocuring process. However, the authors do not explain why this agent was chosen over other silanization methods or agents.
4.    Include a comparative examination of alternative scaffolding materials.
5.    To increase clarity and interpretability, include statistical analysis when presenting data.
6.    Summarise the materials and methods section into a table.
7.    write the full form of phr in line 165,
8.    Add the full printing parameters to the section 2.5 (Formulation, photocuring and 3D printing)
9.    The paper mentions statistical analyses, but detailed results (e.g., p-values for significant findings) would provide clarity. For instance, while antibacterial activity is described as "not statistically significant," quantifying this with precise data could clarify the discussion.
10. Line 90-91, There are incomplete sentences and abrupt transitions that hinder readability. For example, "INSERISCI REF" is an unedited placeholder that disrupts the flow and indicates missing citations.
11. Line 347-349, The effect of methacrylic groups interfering with photoinitiators is mentioned but not substantiated with experimental evidence or literature comparisons.
12. The bioactivity assessment confirms hydroxyapatite (HAp) formation but does not discuss how these results are backed by the literature. No referencing
13. Experimental confirmation would be beneficial for BG-Te-Sil's decreased antibacterial activity caused by silanization.
14. Some sections could benefit from improved organization and clarity, they could be divided into subsections and subsections with distinct titles.
15. Although the report mentions some limitations (such as slower cure rates for filled formulations), it does not specifically identify them. A balanced viewpoint would be achieved by adding a few more points about limitations and future directions
Â
Conclusion
Overall, this study makes a significant addition to bioengineering and material science by providing new insights into the production of functional and durable scaffolds. With some changes to solve the discovered flaws, it might considerably advance the use of 3D-printed scaffolds in hard tissue regeneration and infection-resistant medical devices.
Â
Comments on the Quality of English LanguageIt wouod benefit from proofreading
Author Response
The answer for the reviewer comments are reported in the attached file.Â

Reviewer 2 Report
Comments and Suggestions for Authors
The manuscript with the title "3D Printed Acrylated Soybean Oil Scaffolds with Vitrimeric Properties Reinforced by Tellurium Doped Bioactive Glass"
Manuscript ID: polymers-3368214
Â
Dear Authors,
In the beginning, I would like to express words of my appreciation for the idea and effort put into conducting research and writing the manuscript recommended to me for review. Â
Â
General Comments:Â
The innovative research on the photocurable resins based on acrylated epoxidized soybean oil (AESO) reinforced with tellurium-doped bioactive glass (BG-Te) is presented in the article. The research investigates 3D-printed materials with vitrimers properties, biocompatibility and anti-bacterial actions. This work brings together dynamic covalent polymer networks and sustainable biobased materials as a first to highlight the potential of these materials for hard tissue regeneration and the engendering of infection-resistant medical devices.
In my opinion the research introduces a unique method for designing biobased materials with dynamic covalent properties. Moreover, the article thoroughly examines mechanical, thermal, cytocompatibility, and antibacterial properties.Â
Specific Comments:Â
The work is appropriate for the journal, but there are some observations that must be addressed before the manuscript can be accepted.
1) the description of the fabrication and modification processes for BG-Te-Sil could be expanded with technical details to facilitate reproducibility by other researchers
2)A more detailed analysis of the antibacterial mechanisms, particularly the differences between BG-Te and BG-Te-Sil, would add value
3)Adding a comparative section to evaluate the proposed material against other materials used in tissue engineering would enhance the discussion
Â
Final State of the review:Â
The work presented for review shows the exciting work of authors that was put into this manuscript. The article meets the criteria set for authors to be published in Polymers.Â
Therefore, I ask the Editor to recommend this work for publication after MINOR REVISION.Â
Â
Â
Author Response
The answer to the reviewer comments are reported in the attached file
